# Traditional herbal medicine use doubled the risk of multi-organ dysfunction syndrome in children: A prospective cohort study

**Nahom Worku Teshager**[1]*, **Ashenafi Tazebew Amare**[1], **Koku Sisay Tamirat**[2], **Mulualem Endeshaw Zeleke**[3], **Asefa Adimasu Taddese**[2]

**1** Department of Pediatrics and Child Health, School of Medicine, College of Medicine and Health Sciences, University of Gondar, Gondar, Ethiopia, **2** Department of Epidemiology and Biostatistics, Institute of Public Health, College of Medicine and Health Sciences, University of Gondar, Gondar, Ethiopia, **3** Department of Anesthesia, School of Medicine, College of Medicine and Health Sciences, University of Gondar, Gondar, Ethiopia

* nahomarm@gmail.com

**Data Availability Statement:** Data is submitted as a supporting information file.

**Funding:** The study was funded by the University of Gondar, and the funder had no role in study

## Abstract

### Background

Traditional herbal medicine (THM) is frequently used in pediatric populations in many low-income countries as a form of healthcare and has been associated with a range of adverse events, including liver toxicity, renal failure, and allergic reactions. Despite these concerns, its impact on multi-organ dysfunction syndrome (MODS) risk has not been thoroughly investigated.

### Objective

This study aimed to investigate the incidence and predictors of MODS in a pediatric intensive care unit (PICU) in Ethiopia, with a focus on the association between THM use and the risk of MODS.

### Methods

This was a single-center prospective cohort study conducted at a PICU in the university of Gondar Comprehensive Specialized hospital, Northwest Ethiopia. The study enrolled eligible patients aged one month to 18 years admitted to the PICU during the study period. Data on demographic characteristics, medical history, clinical and laboratory data, and outcome measures using standard case record forms, physical examination, and patient document reviews. The predictors of MODS were assessed using Cox proportional hazards models, with a focus on the association between traditional herbal medicine use and the risk of MODS.

### Results

A total of 310 patients were included in the final analysis, with a median age of 48 months and a male-to-female ratio of 1.5:1. The proportion and incidence of MODS were 30.96%

design, data collection, analysis, interpretation, decision to publish, or manuscript preparation. https://uog.edu.et/.

**Competing interests:** There were no competing interests, and patients/public were not involved in the research.

(95% CI:25.8, 36.6) and 7.71(95% CI: 6.10, 9.40) per 100-person-day observation respectively. Renal failure (17.74%), neurologic failure (15.16%), and heart failure (14.52%) were the leading organ failures identified. Nearly one-third of patients (32.9%) died in the PICU, of which 59.8% had MODS. The rate of mortality was higher in patients with MODS than in those without. The Cox proportional hazards model identified renal disease (AHR = 6.32 (95%CI: 3.17,12.61)), intake of traditional herbal medication (AHR = 2.45, 95% CI:1.29,4.65), modified Pediatric Index of Mortality 2 (mPIM 2) score (AHR = 1.54 (95% CI: 1.38,1.71), and critical illness diagnoses (AHR = 2.68 (95% CI: 1.77,4.07)) as predictors of MODS.

## Conclusion

The incidence of MODS was high. Renal disease, THM use, mPIM 2 scores, and critical illness diagnoses were independent predictors of MODS. A more than twofold increase in the risk of MODS was seen in patients who used TMH. Healthcare providers should be aware of risks associated with THM, and educate caregivers about the potential harms of these products. Future studies with larger sample sizes and more comprehensive outcome measures are needed.

## Introduction

Multiorgan dysfunction syndrome (MODS) is a serious condition that can occur in critically ill children, and it is characterized by the gradual failure of two or more organ systems. MODS can arise from a range of underlying conditions and is associated with high morbidity and mortality rates [1]. In pediatric intensive care units (PICUs), the proportion of cases of MODS has been reported to range from 5% to 50% [2–4]. MODS has been found to contribute significantly to an increase in hospital mortality [2, 3, 5], longer hospital stays [6], increased rates of mechanical ventilation [7], and poor health-related quality of life [8].

Low-income countries, including Ethiopia, have a high burden of infectious diseases, malnutrition, and limited healthcare resources [9, 10]. Several factors have been identified as increasing the risk of MODS in children, including age [11, 12], baseline severity of illness [13], comorbid conditions [14], and sepsis [15]. However, there is a lack of research on MODS in low-income settings, and the incidence and predictors of MODS in the pediatric population in Ethiopia remain poorly understood. Identifying the incidence and predictors of MODS is crucial for identifying high-risk patients and designing appropriate interventions.

Traditional herbal medicine (THM) use is a widespread and popular form of healthcare in many low-income countries, including Ethiopia. However, the safety and efficacy of THM are often unclear, and there are concerns about the potential for adverse effects. THM use has been associated with a range of adverse events in children, including liver toxicity, renal failure, and allergic reactions [16–18]. Despite these concerns, THM is frequently used in pediatric populations in low-income countries, and its impact on MODS risk has not been thoroughly investigated [19].

Therefore, this prospective cohort study aims to determine the incidence and predictors of MODS in a PICU in Ethiopia, with a particular focus on the impact of THM use on MODS. The hypothesis is that THM use will increase the risk of MODS in critically ill children. The

results of this study will provide valuable insights into the incidence and predictors of MODS in Ethiopia and will help guide clinical decision-making and improve patient outcomes.

## Materials and method

### Study design, period and setting

This is a single-center prospective cohort study conducted from February 1, 2018, to July 30, 2019, at the Pediatric Intensive Care Unit (PICU) of the University of Gondar Comprehensive Specialized Hospital, Ethiopia. The hospital serves for more than 11 million people with a total of 1000 beds and 195 beds in the pediatrics side, where a multidisciplinary team of professionals provide a range of inpatient and outpatient health care services for approximately 15,000 children beyond the neonatal age coming from the northwest part of the country each year. Major causes of pediatric admissions are pneumonia, malaria, neonatal infections, tuberculosis, meningitis, multiorgan failure and other various types of metabolic and organ system-based emergencies.

The PICU has six beds equipped with electronic monitors and one mechanical ventilator, and the monthly pediatric critical care admissions range from 20 to 30. However, the PICU lacks a pediatric intensivist, respiratory therapist, pharmacist, and dietician, and the team composition usually consists of a general pediatrician, resident, interns, and a few senior-level nurses.

### Population and sample

The study recruited all eligible participants aged 1 month to 18 years who were admitted to the PICU during the study period. Surgical patients were admitted for recovery purposes only and those with incomplete data were excluded from the study. The sample size was determined using the sample size formula for incidence proportion with finite population correction for a population of less than 10,000 (In this case, N = 450 as nearly 25 patients are admitted monthly in the hospital), and 10% contingency. With an expected incidence proportion of 57%(P) for MODS, a margin of error of 5%(e), a 95% confidence level(Z), and an 18-month (1.5 year) follow-up period(T) a sample size of 566 patients was calculated using the formula: $n_0 = Z^2 * P * (1-P) * T / e^2$. Upon correction of this figure for a finite population using the formula: nf = n0/ 1+ (n0/N) and 10% contingency added, the final sample size was found to be 276 patients.

Data were collected from 327 patients who fulfilled the inclusion criteria using a consecutive sampling technique. After excluding 14 patients with incomplete data on the outcome variable, and three patients with extreme outlier values, a total of 310 patients were included in the final analysis.

### Data collection procedure

Data were collected by treating physicians using standard data collection forms after taking parental consent. Sociodemographic and medical history data were obtained by interviews, while diagnosis, laboratory indices, and clinical course data were collected through chart reviews at discharge. The severity of illness at admission was assessed using the Pediatric Index of Mortality (PIM 2) score, and only primary diagnoses were used for the WHO international classification of Diseases (ICD-10) assignment in patients with multiple diagnoses. Clinical parameters needed for PIM 2 scoring were recorded at admission. The data were verified by the data collector and principal investigator, who also provided regular orientations and training every three months and demonstrations every Monday as per the rotation of physicians in the service area. The principal investigator supervised the process and checked the

completeness of case record forms daily, while investigators only accessed patient records and did not provide direct patient care. Ethical clearance was obtained from the Ethical Committee of the school of Medicine at the University of Gondar's College of Medicine and Health Sciences, and informed verbal consent was obtained from caregivers.

## Variable of the study and operational definitions

The main outcome variable of the study was the time to development of MODS (event). The independent variables considered were sociodemographic factors such as age, sex, caregiver's educational status, occupation, and caregivers, as well as clinical characteristics such as traditional herbal medication intake, duration of illness, source of admission, diagnosis, critical illness diagnosis, comorbidity, nutritional status, vaccination status, interventions administered in the PICU, and pre-admission treatments (including fluid resuscitation), mPIM 2 score, complications, and treatment outcome.

Event (MODS): In this study, MODS is defined as having physiologic derangement in two or more organ systems.

Censored: refers to those patients with no event of interest.

Length of stay (LOS): refers to the duration of stay in days from the date of admission to the date of discharge.

Short-term outcome: the outcome of the patient until he or she leaves the hospital including death.

Critical illness: refers to sepsis, severe sepsis, or septic shock within 24 hours of admission or acute respiratory distress syndrome during PICU admission.

## Data processing and analysis

After ensuring consistency and completeness, the data was imported into EpiData V.3.1, then exported to Excel and subsequently to R 4.2.2 for cleaning and analysis. Descriptive statistics such as mean, median, and proportions were used to summarize the baseline characteristics and admission patterns of the study population. Additionally, summary statistics such as life-table, log-rank test, and Kaplan-Meier curves were computed to determine the incidence rate of MODS and compare survival curves between different categories of explanatory variables. Finally, both bivariate and multivariable Cox proportional hazards models were used to identify the predictors of MODS. Variables with a p-value of less than 0.2 in the bivariate analysis were included in the multivariable proportional hazard model. A 95% confidence intervals of hazard ratios were computed, and variables with a p-value of less than 0.05 in the multivariate Cox proportional hazards model were considered significantly and independently associated with the dependent variable. To ensure the fitness of the Cox proportional hazards model, the Schoenfeld residuals test was performed.

## Result

### Sociodemographic characteristics

The final analysis of the study included 310 patients. The majority of patients were young children, with a median age at admission of 48 months (interquartile range: 12–126). The male-to-female ratio was 1.5:1, indicating a slight preponderance of males in the sample.

The majority of caregivers (92.9%) were parents, with grandparents, siblings, and others making up a small proportion. In terms of their level of education, 77.7% of caregivers had no formal education, while 10.3%, 5.5%, and 6.5% had primary, secondary, and college/university education, respectively.

Regarding the caregivers' occupations, the majority of them were farmers (71.3%), followed by merchants (10%), government employees (10%), daily laborers (8.1%), and unemployed individuals (0.6%). In terms of the patient's residences, the majority (77.1%) lived in rural areas, with the remaining 22.9% residing in urban areas (Table 1).

These findings suggest that the study population was predominantly rural with a low level of education among caregivers and a significant proportion of patients suffering from malnutrition. However, it is important to note that these characteristics may not be representative of other populations or settings, and further research would be needed to confirm the generalizability of these findings.

## The clinical condition of patients

The primary source of admissions to the PICU was the emergency room, accounting for 83.9% of all admissions. The median duration of illness before admission to the PICU was 7 days (interquartile range: 3–10). Neurologic (22.9%), Infectious (18.39%), and trauma and environmental emergencies (10.97%) were the leading causes of admission.

Respiratory failure was the leading cause of admission to the PICU, accounting for 19.0% of cases, followed by septic shock and increased intracranial pressure, both accounting for 14.3% of cases.

Nearly one-third (32.3%) had a critical illness diagnosis at admission, of which 32% had sepsis, 9% severe sepsis, 47% septic shock, and the remaining (12%) had acute respiratory distress syndrome. More than half (55.2%) of patients had comorbid illness. The median Pediatric Index of Mortality 2 (PIM 2) score was -3.38 (interquartile range: -4.61, -2.29) (Table 1 and Fig 1).

## Multi-organ Dysfunction syndrome (MODS) among ICU patients

The study monitored 310 ICU patients for a total of 1245 person-day observations (41.5 person-months) and found that 30.96% (95% CI: 25.8, 36.6) developed Multi-Organ Dysfunction Syndrome (MODS). Among those with MODS, 42.70% had it at admission or within 24 hours, while the remainder developed it during their ICU stay, resulting in an incidence rate of 7.71 (95% CI: 6.10, 9.40) per 100-person day observation. The median time to the development of MODS was 11.81 days (Fig 2).

The most common organ failures among the study participants (n = 310) were renal failure (17.74%), neurologic failure (15.16%), and heart failure (14.52%) (Fig 3). Of the patients studied, 32.9% died in the PICU, and 59.8% of those who died had MODS. Patients with MODS had seven-fold higher odds of mortality ($\chi^2$ = 7.35, df = 1, p = 0.001) compared to those without MODS. The length of hospital stay was reduced by 20% for patients with MODS ($R^2$ = 0.22, β = -1.20, p = 0.009), potentially due to early mortality. However, there was no significant difference in mechanical ventilation between the two groups ($\chi^2$ = 1.94, df = 1, p = 0.083).

## Predictors of MODS in the PICU

The study utilized Cox proportional hazards model to identify predictors of MODS in the PICU. The multivariate analysis identified four predictors of MODS in patients with MODS: renal diseases, intake of herbal medication before admission, PIM 2 score, and presence of critical illness diagnoses. Patients with renal diseases had a hazard of developing MODS 6.32 times higher (AHR = 6.32, 95%CI: 3.17 to 12.61) compared to those without renal disease, while those who took herbal medication had a hazard of 2.45 times higher risk of developing MODS (AHR = 2.45, 95%CI: 1.29 to 4.65). Additionally, patients who had any of the critical

**Table 1. Sociodemographic and clinical characteristics of patients (N = 310).**

| Characteristics | Frequency | Percentage (%) |
|---|---|---|
| **Sociodemographic Characteristics** | | |
| **Age** | | |
| Infant | 88 | 28.4 |
| Toddler | 29 | 9.4 |
| Preschools | 63 | 20.3 |
| School age | 63 | 20.3 |
| Adolescent | 67 | 21.6 |
| **Sex** | | |
| Male | 184 | 59.4 |
| **Residence** | | |
| Urban | 71 | 22.9 |
| Rural | 239 | 77.1 |
| **Care Givers** | | |
| Parents | 288 | 92.9 |
| Grand parents | 8 | 2.6 |
| Siblings | 8 | 2.6 |
| Others | 6 | 1.9 |
| **Care-givers' Level of Education** | | |
| No formal education | 241 | 77.7 |
| Primary school | 32 | 10.3 |
| Secondary school | 17 | 5.5 |
| College or university | 20 | 6.5 |
| **Care-givers' Occupation** | | |
| Farmers | 221 | 71.3 |
| Merchants | 31 | 10.0 |
| Government employee | 31 | 10.0 |
| Daily laborer | 25 | 8.1 |
| Unemployed | 2 | .6 |
| **Clinical characteristics** | | |
| **Sources of Admission** | | |
| Emergency Room | 260 | 83.9 |
| Wards | 50 | 16.1 |
| **Comorbid Illness** | | |
| Yes | 139 | 44.8 |
| No | 171 | 55.2 |
| **Nutritional Status** | | |
| Normal | 149 | 48.1 |
| Wasted | 161 | 51.9 |
| **Vaccination Status** | | |
| Complete | 202 | 65.2 |
| Incomplete | 108 | 34.8 |
| **Critical Illness Diagnosis** | | |
| Sepsis within 24 hours | 32 | 10.3 |
| Sever Sepsis within 24 hours | 9 | 2.9 |
| Septic Shock within 24 hours | 47 | 15.2 |
| ARDS during Admission | 12 | 3.9 |
| **Inotropes** | | |

*(Continued)*

**Table 1.** (Continued)

| Characteristics | Frequency | Percentage (%) |
|---|---|---|
| Yes | 59 | 19.0 |
| No | 251 | 81.0 |
| **Mechanical Ventilation** | | |
| Yes | 36 | 11.6 |
| No | 274 | 88.4 |
| **Complications** | | |
| Yes | 253 | 81.6 |
| No | 57 | 18.4 |

ARDS: Acute Respiratory Distress Syndrome

illness diagnoses had a hazard of developing MODS 2.68 times higher (AHR = 2.68, 95%CI: 1.77 to 4.07) (Fig 4), and every one-unit increase in the modified PIM 2 score increased the hazard of developing MODS 1.54 times, keeping other variables constant (AHR = 1.54, 95% CI: 1.38 to 1.71). These findings suggest that early recognition and management of these predictors may improve patient outcomes in the PICU (Table 2).

## Discussion

The prospective cohort study conducted in a pediatric intensive care unit (PICU) in Ethiopia aimed to investigate the incidence of multiorgan dysfunction syndrome (MODS) and its potential predictors, with a particular focus on traditional herbal medicine use. The study

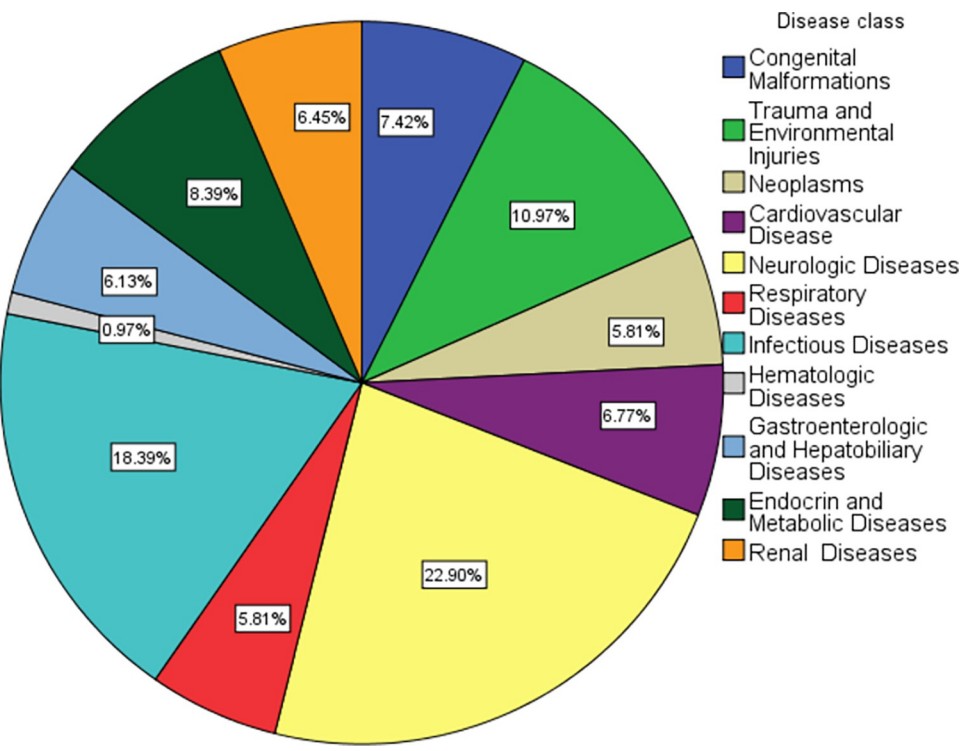

**Fig 1. Diagnoses of patients according to the ICD-10 classification (n = 310).**

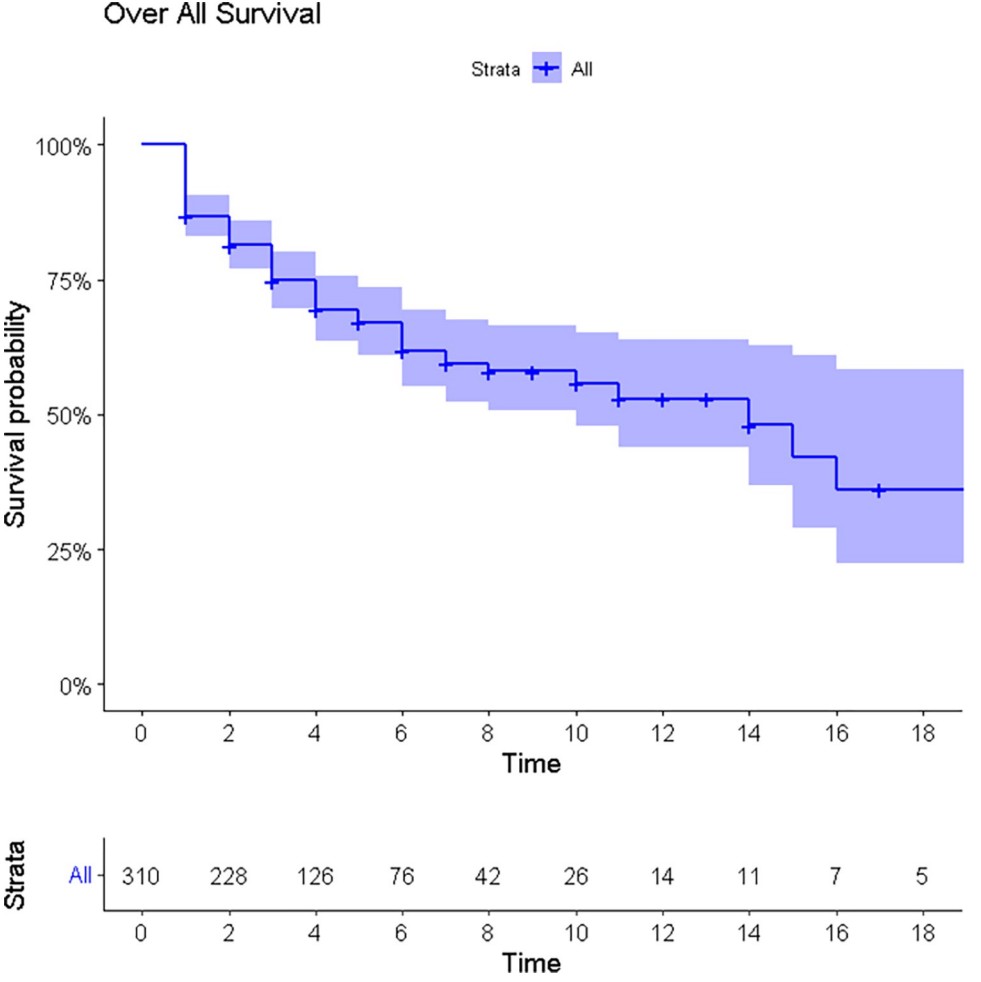

**Fig 2. Kaplan Meier estimate curve for overall survival of patients.**

revealed that the incidence of MODS in the PICU was 7.71 per 100-person day observation, with a proportion of 30.96%. This incidence was lower than that reported in a prospective follow-up study in Peru [3] but higher than the reports from several other studies [8, 20–22]. The difference in patient characteristics, disease severity, and healthcare resources availability across settings could explain the differences in the proportion and incidence rates of MODS.

The study findings showed that patients with renal disease had a hazard of developing MODS that was 6.32 times higher compared to those without renal disease. Renal dysfunction can lead to the accumulation of metabolic waste products and electrolyte imbalances, which can cause systemic inflammations and organ dysfunction, contributing to or failures in multiple systems [23–25]. Additionally, renal disease may serve as a marker of overall disease severity or susceptibility to other risk factors for MODS, as supported by multiple literatures [26–33].

The findings of the study suggest that the use of traditional herbal medicines by critically ill children significantly increases the risk of developing multiorgan dysfunction syndrome (MODS). The risk was found to be twice higher in children who took traditional herbal medicines compared to those who did not. This finding is consistent with previous studies [34–38] conducted in both low- and high-income settings, indicating that the risk of MODS associated

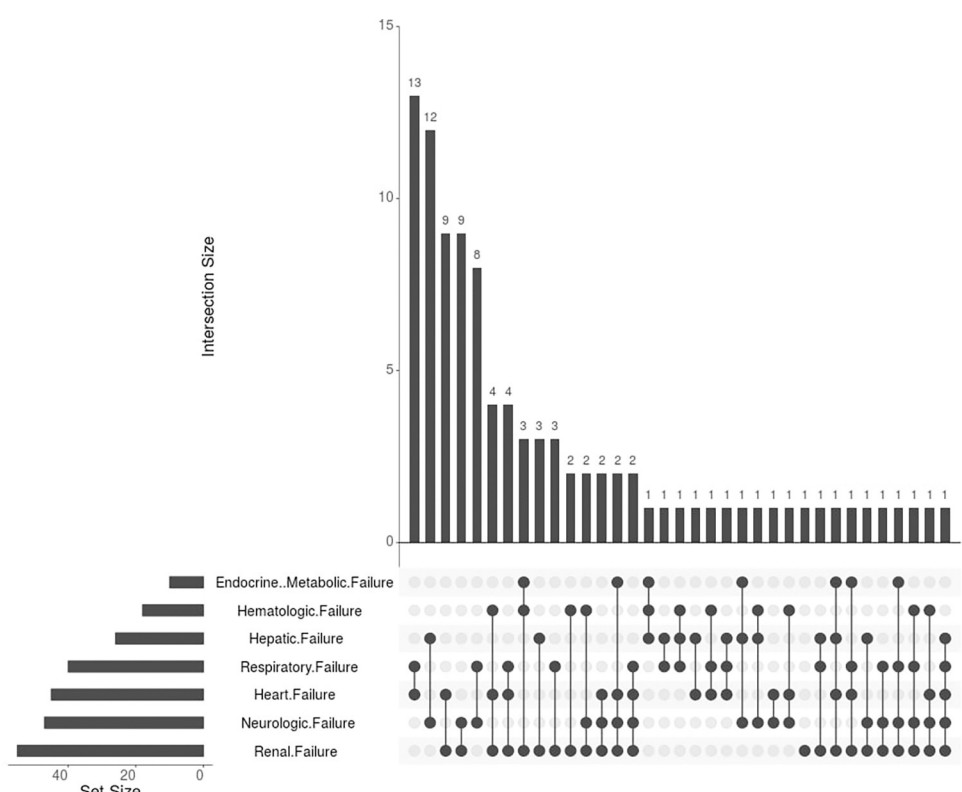

**Fig 3. Pattern of organ dysfunction in patients with MODS (n = 96).**

with traditional herbal medicine use is not limited to a specific geographical region or economic status. There could be several reasons for the increased risk of MODS in critically ill children who take traditional herbal medicines. Firstly, herbal medicines may interact with other medications or treatments that the child is receiving, leading to adverse reactions and potentially increasing the risk of MODS [37, 39]. Secondly, children who take herbal medicines may have underlying health conditions that are not adequately treated with conventional medicine, which increases their risk of developing MODS [34, 40–42]. It is possible that the use of herbal medicines may have delayed or interfered with proper medical treatment, leading to worse health outcomes [43]. In addition, traditional herbal medicines are often given or adulterated with drugs, food-related or non-food additives, such as local alcohol beverages like Tela or Areqi, which may have contributed to the toxic effects of the herbal medicines [44–46]. These additives may have harmful interactions with the herbal medicines, further exacerbating the risk of adverse health outcomes. Furthermore, the lack of standardization and quality control in the production of traditional herbal medicines may have contributed to the increased risk of adverse health outcomes [16, 47, 48]. The inherent toxic effects and high dosages of some herbal medicines can also be a factor that contributes to the increased risk of MODS. Overall, the findings of this study highlight the need for caution when using traditional herbal medicines, particularly in critically ill children. There is a need for more standardized production and quality control measures to ensure the safety and efficacy of traditional herbal medicines. It is important to educate healthcare providers and the public on the potential risks associated with herbal medicine use, and to encourage the integration of traditional medicine with modern medical practices to provide the best possible care for critically ill children.

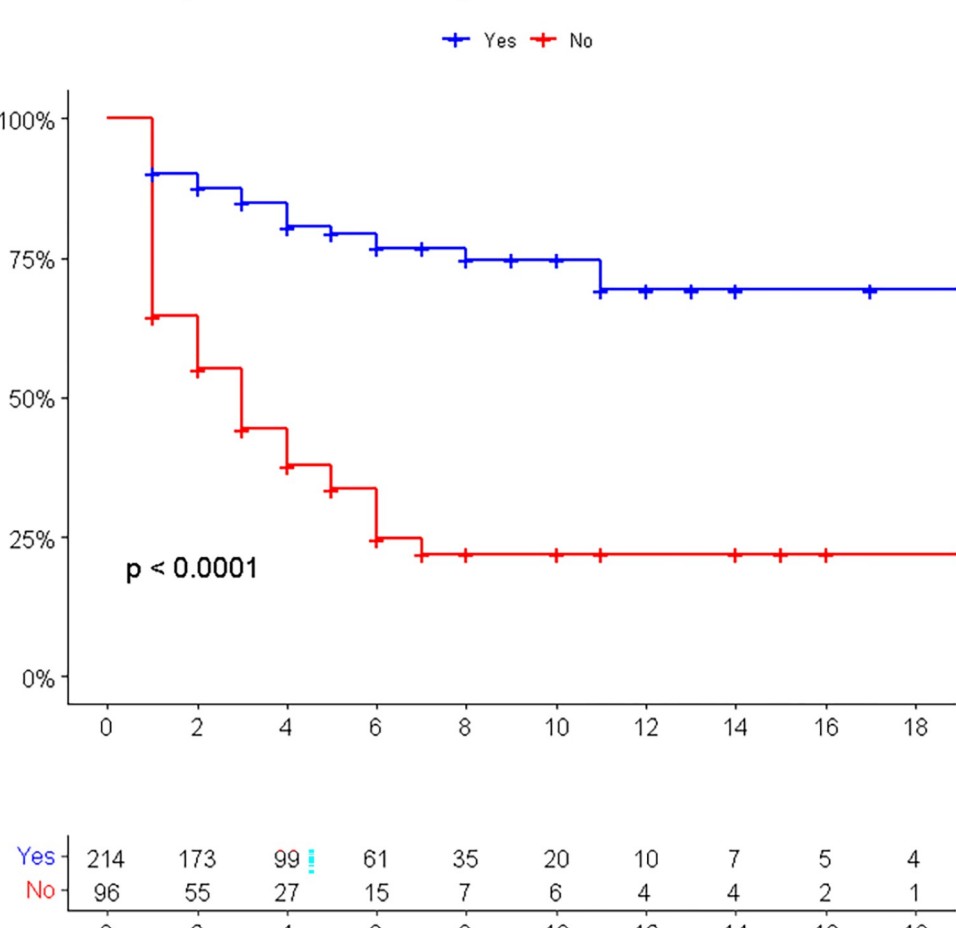

**Fig 4. Kaplan Meier estimate curve for critical illness diagnosis as a predictor of MODS.**

The finding that patients with critical illness diagnoses had a 2.68 times increased risk of developing MODS in critically ill patients is supported by multiple other studies. Critical illness diagnoses, such as sepsis, septic shock, and ARDS, have been associated with increased risk of MODS in pediatric patients [2, 49, 50]. These studies reported that sepsis, which is among the types of critical illness diagnoses, was the most common primary diagnosis in critically ill children with MODS. Another study also showed that septic shock was the most common cause of MODS in the PICU [51]. This finding could be due to the systemic inflammatory response and reduced blood flow to vital organs that often accompany these conditions. In addition, the use of certain medications and treatments in critically ill children, such as mechanical ventilation or vasoactive agents, may also contribute to the development of MODS.

The study also used the modified Pediatric Index of Mortality 2 (PIM2) score to assess the baseline severity of illness of patients at admission to the PICU [52]. The PIM2 score has been shown to be a useful predictor of MODS in previous studies [53, 54]. The modified PIM2 score included SpO2 as a substitute for PaO2 and anion gap due to the unavailability of an arterial blood gas analyzer in the PICU during the study period [13, 52]. The finding that a one-unit increase in the modified PIM2 score increases the hazard of developing MODS 1.54

**Table 2. Bivariable and multivariable Cox PH model for explanatory variables.**

| Variables | Multiorgan Dysfunction Syndrome | | CHR (95%CI) | AHR (95%CI) | P-Value |
|---|---|---|---|---|---|
| | Yes | No | | | |
| **Age in month** | | | | | |
| Infant | 26 | 62 | 1 | 1 | |
| Toddler | 10 | 19 | 1.03(0.49, 2.15) | 1.87(0.86,4.03) | 0.11 |
| Preschools | 21 | 42 | 1.26(0.70,2.25) | 1.29(0.69,2.41) | 0.41 |
| School age | 23 | 40 | 1.36(0.77,2.39) | 1.81(0.98,3.33) | 0.06 |
| Adolescent | 16 | 51 | .94(0.50,1.77) | 1.54(0.78,3.03) | 0.21 |
| **Sex** | | | | | |
| Male | 56 | 128 | 1 | 1 | |
| Female | 40 | 86 | 1.04(0.69,1.57) | 0.89(0.58,1.36) | 0.59 |
| **Address** | | | | | |
| Urban | 17 | 54 | 1 | 1 | |
| Rural | 79 | 160 | 1.29(0.76,2.18) | 1.42(0.83,2.44) | 0.19 |
| **Time of admission** | | | | | |
| Day shift | 48 | 133 | 1 | | |
| Night shift | 48 | 81 | 1.54(1.03,2.30) | 1.21(0.79,1.84) | 0.36 |
| **Source of admission** | | | | | |
| Emergency Room | 72 | 188 | 1 | 1 | |
| Wards | 24 | 26 | 2.20(1.38,3.51) | 1.16(0.64,2.11) | 0.62 |
| **mPIM 2 score** | -3.21±1.82 | | 1.47(1.33,1.63) | 1.54(1.38,1.71) | 0.001 |
| **Duration of illness** | 7.00 ± 11.57 | | 1.02(1.00, 1.03) | 1.01(0.99,1.03) | 0.07 |
| **Renal Disease** | | | | | |
| Yes | 11 | 9 | 2.75(1.45,5.22) | 6.32(3.17,12.61) | 0.001 |
| No | 85 | 205 | 1 | 1 | |
| **Critical illness Diagnosis** | | | | | |
| Yes | 52 | 48 | 2.73(1.82,4.10) | 2.68(1.77,4.07) | 0.001 |
| No | 44 | 166 | 1 | 1 | |
| **Herbal Medication** | | | | | |
| Yes | 11 | 12 | 1.65(0.88,3.11) | 2.45(1.29,4.65) | 0.001 |
| No | 85 | 202 | 1 | 1 | |
| **Use of Inotropes** | | | | | |
| Yes | 36 | 23 | 3.13(2.07,4.75) | 1.65(0.99,2.77) | 0.054 |
| No | 60 | 191 | 1 | 1 | |
| **Complications** | | | | | |
| Yes | 91 | 162 | 4.11(1.67,10.13) | 2.42(0.96,6.04) | 0.058 |
| No | 5 | 52 | 1 | 1 | |
| **Length of Hospital Stay** | 10.79±12.90 | | 0.97(0.94,0.99) | 0.98(0.96,1.00) | 0.10 |

AHR: Adjusted Hazard Ratio, CHR: Crude Hazard Ratio, mPIM2: Modified Pediatric Index of Mortality 2score

times in critically ill children may be explained by the inclusion of clinical and physiological parameters related to organ dysfunction as well as respiratory and cardiovascular function in the scoring system.

## Strength and limitations of the study

One of the strengths of this study is that it utilized a prospective cohort design, which allowed for the collection of data over a period of time, and therefore, helped to establish the temporal

relationship between traditional herbal medicine use and the development of MODS in critically ill children. Additionally, the use of survival analysis, a statistical technique that allowed for the estimation of the probability of developing MODS over time, is a robust method for analyzing longitudinal data.

However, this study also has some limitations. Firstly, it was conducted at a single center with a small sample size, which may limit the generalizability of the findings to other settings. Moreover, the study did not examine the specific types of THM which might have different risk and benefits. Furthermore, as data collection was limited to admission and discharge, important predictors of MODS that may have developed during hospitalization may have been missed.

Another limitation is the use of MODS as the sole outcome measure, which may not fully capture the overall outcomes of critically ill children in the PICU. Additionally, the study did not fully evaluate the impact of the availability of trained staff and equipment for diagnosis, treatment, and monitoring on the incidence and outcome of MODS. Lastly, the modified PIM 2 score, while a useful predictor of MODS, was based on only 9 parameters, which may have limited its accuracy.

In summary, while this study provides important insights into the potential risks associated with traditional herbal medicine use in critically ill children, its limitations should be considered when interpreting the findings. Further research with larger sample sizes, more comprehensive data collection, and a focus on evaluating the impact of healthcare infrastructure on patient outcomes is needed to better understand the risks associated with traditional herbal medicine use in the PICU.

## Conclusion

In conclusion, this prospective cohort study highlights the significant risk of MODS in critically ill children admitted to a PICU in Ethiopia. The study found that traditional herbal medicine use significantly increased the risk of developing MODS by nearly three times, indicating the need for healthcare providers to be cautious of such practices. The modified PIM 2 score, including parameters related to organ dysfunction, respiratory and cardiovascular function, was also found to be a useful predictor of MODS in critically ill children. The study's limitations include its single-center design with a small sample size and a limited outcome measure of MODS. Healthcare providers should consider the potential predictors identified in this study for early recognition and management of MODS to improve outcomes. Further studies with larger sample sizes and more comprehensive outcome measures are necessary to enhance care for critically ill children in Ethiopia and beyond.

## Supporting information

**S1 Data.**
(XLS)

## Acknowledgments

We express our gratitude to the study participants, data collectors, supervisors, and hospital administrators of the University of Gondar comprehensive specialized hospital, as well as the department of pediatrics and child health.

## Author Contributions

**Conceptualization:** Nahom Worku Teshager, Ashenafi Tazebew Amare, Koku Sisay Tamirat.

**Data curation:** Nahom Worku Teshager.

**Formal analysis:** Nahom Worku Teshager, Koku Sisay Tamirat, Mulualem Endeshaw Zeleke.

**Funding acquisition:** Nahom Worku Teshager.

**Investigation:** Nahom Worku Teshager.

**Methodology:** Nahom Worku Teshager, Ashenafi Tazebew Amare.

**Project administration:** Nahom Worku Teshager.

**Resources:** Nahom Worku Teshager.

**Software:** Nahom Worku Teshager, Mulualem Endeshaw Zeleke.

**Supervision:** Nahom Worku Teshager, Mulualem Endeshaw Zeleke, Asefa Adimasu Taddese.

**Validation:** Nahom Worku Teshager, Ashenafi Tazebew Amare, Koku Sisay Tamirat, Mulualem Endeshaw Zeleke, Asefa Adimasu Taddese.

**Visualization:** Nahom Worku Teshager, Mulualem Endeshaw Zeleke, Asefa Adimasu Taddese.

**Writing – original draft:** Nahom Worku Teshager, Asefa Adimasu Taddese.

**Writing – review & editing:** Nahom Worku Teshager, Ashenafi Tazebew Amare, Koku Sisay Tamirat, Mulualem Endeshaw Zeleke, Asefa Adimasu Taddese.

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
