## [Decision Letter · Decision Letter 0]

6 Sep 2023

PONE-D-23-14246Traditional Herbal Medicine Use Tripled the Risk of Multi-organ Dysfunction Syndrome in a Pediatric Intensive Care Unit in Ethiopia: A Prospective Cohort Study.PLOS ONE

Dear Dr. Teshager,

Thank you for submitting your manuscript to PLOS ONE. After careful consideration, we feel that it has merit but does not fully meet PLOS ONE’s publication criteria as it currently stands. Therefore, we invite you to submit a revised version of the manuscript that addresses the points raised during the review process.

The manuscript has been evaluated by two reviewers, and their comments are available below. The reviewers have raised a number of major concerns. They feel that the limitations of the manuscript have not been sufficiently discussed. Given the limitations of the study design, care must be taken to ensure that conclusions - including the title - are presented appropriately and not overstated. The reviewers also note concerns about the reporting of the Methods and Results, including statistical analyses presented, and in particular they request clarifications regarding the sample size calculation.

Could you please carefully revise the manuscript to address all comments raised?

We look forward to receiving your revised manuscript.

Kind regards,

Marianne Clemence

Staff Editor

PLOS ONE

Journal Requirements:

4. PLOS requires an ORCID iD for the corresponding author in Editorial Manager on papers submitted after December 6th, 2016. Please ensure that you have an ORCID iD and that it is validated in Editorial Manager. To do this, go to ‘Update my Information’ (in the upper left-hand corner of the main menu), and click on the Fetch/Validate link next to the ORCID field. This will take you to the ORCID site and allow you to create a new iD or authenticate a pre-existing iD in Editorial Manager. Please see the following video for instructions on linking an ORCID iD to your Editorial Manager account: https://www.youtube.com/watch?v=_xcclfuvtxQ.

5. Please ensure that you refer to Figures 3, 4 and 5 in your text as, if accepted, production will need this reference to link the reader to the figure.

Reviewers' comments:

Reviewer's Responses to Questions

**Comments to the Author**

1. Is the manuscript technically sound, and do the data support the conclusions?

Reviewer #1: Yes

Reviewer #2: Yes

2. Has the statistical analysis been performed appropriately and rigorously? 

Reviewer #1: Yes

Reviewer #2: Yes

3. Have the authors made all data underlying the findings in their manuscript fully available?

Reviewer #1: Yes

Reviewer #2: Yes

4. Is the manuscript presented in an intelligible fashion and written in standard English?

Reviewer #1: Yes

Reviewer #2: Yes

5. Review Comments to the Author

Reviewer #1: I found it very interesting topic and it is thoroughly studied with an exhaustively analyzed data. If the paper gets published many of the public health experts will work on the mitigation strategy through public awareness. However I have some comments

1. The title seems exaggerated. Although the patients who used THM had a 2.45 (95% CI: 1.29, 4.65 ) risk of developing MODS, the authors are trying to exaggerate the output. Hence it has to be said doubled the occurrence of MODS instead of a threefold increase in risk of MODS.

2. In the abstract, under the result section, the last sentence is repeatedly mentioned and has to be deleted. The word keyword has to be written as one word and has to be corrected.

3. In the material and methods section, the study area description is needed. Information regarding the area coverage of the hospital, patient types attending the hospital has to be provided.

4. In the calculation of the sample size, the formula yields 376 not 260. If 10% contingency is used it will be 414. Rather, you can estimate the number of admissions based on a three month data and then you can use a sample adjustment formula for finite population.

5. In the discussion section (paragraph 2, line 4), the justification you indicated has to be accompanied by references.

6. In the discussion section paragraph 3, the justification you raised on the increased risk of MODS in critically ill children who take traditional herbal medicines has to be supported by references.

7. In the introduction (on the last paragraph) and data processing and analysis part, avoid using first-person pronouns ("we").

Reviewer #2: summary of the research:

The study provides important insights into the incidence and predictors of multi-organ dysfunction syndrome (MODS) in a pediatric intensive care unit (PICU) in Ethiopia. The finding that traditional herbal medicine (THM) use is associated with a threefold increased risk of MODS is particularly concerning, given the widespread use of such medicines in Ethiopia and other low- and middle-income countries. The study's strengths include its rigorous methodology and focus on a vulnerable population, and its findings have important implications for clinical practice and public health policy.

However, there are some limitations to consider. The study was conducted at a single center, which may limit the generalizability of the findings to other settings. The sample size was relatively small, which may have limited the ability to fully explore the association between THM use and MODS. The study did not examine the specific types of THM used, which may have different risks and benefits.

General Comments:

Comments on title and abstract:

1. Title not specific to study area: I will suggest that the authors revise the title to make it more specific to the study area. This could help readers quickly understand what the study is about and whether it is relevant to their interests.

2. Lack of background of the study: I suggest that the authors include more information about the background of the study in the background section of the abstract. This could help readers understand the context of the research and why it is important.

3. It is not explicitly stated in the abstract where the single center study took place. To improve the readers' understanding of the study's limitations and generalizability, I suggest that the authors include a sentence in the abstract specifying that the study was conducted at Gondar Comprehensive Specialized Hospital, Gondar, Ethiopia

The authors should provide a more detailed explanation of the following:

Population and sample part (p.4):

The sample size for the study appears to have been determined using a standard formula for incidence proportion with finite population correction, taking into account the expected incidence proportion of MODS, margin of error, confidence level, and follow-up period. However, there is some discrepancy in the reported sample size calculation, with the statement indicating a sample size of 260 patients, while later indicating that a total of 310 patients were included in the final analysis. Moreover, there is no explicit mention of sampling techniques

Variables of the study part (p.5)

1. There is no explicit mention of a variable related to traditional herbal medicine (THM) use in the list of independent variables considered in the study.

2. The authors of a study on Multiple Organ Dysfunction Syndrome (MODS) should include information about the criteria used for the diagnosis of MODS in their manuscript.

Data processing and analysis (p.6)

The statistical methods used in the study should be reviewed to ensure that appropriate methods were used for the type of data analyzed. For analyzing time-to-event data, such as the time to development of MODS, survival analysis techniques such as bivariate and multivariate Cox proportional hazards models are appropriate, while the chi-square test and simple linear regression are not. Therefore, the chi-square test and simple linear regression should be replaced with appropriate statistical methods like bivariate and multivariate Cox proportional hazards models. However, it appears that the appropriate statistical methods were used in the results section of the manuscript.

Results part:

1. As the objective of the study is to examine the incidence and predictors of Multi-Organ Dysfunction Syndrome (MODS) in a Pediatric Intensive Care Unit (PICU) in Ethiopia, the Cox proportional hazards model can be utilized to investigate the association between predictor variables and the likelihood of developing MODS over time, rather than the mortality of patients (p.10). Mortality of patients should be removed.

2. Patients who used THM had an increased risk of MODS AHR of 2.45 (95% CI: 1.29, 4.65). Therefore, a hazard ratio of 2.45 is slightly lower than a three-fold increase in risk. It will be better nearly three-fold.

Regarding figures:

Figures 3, 4, and 5 are likely to be important components of the study and should be properly described, labeled, and cited in the text to ensure that readers can fully understand and interpret the results presented. It is important that the author reviews the manuscript and ensures that figures are properly integrated into the text to support the study's findings and conclusions.

6. PLOS authors have the option to publish the peer review history of their article (what does this mean?). If published, this will include your full peer review and any attached files.

Reviewer #1: **Yes: **Dr. Dawit Kassaye Getaneh (Assistant Prof. of Epidemiology)

Reviewer #2: No

---

## [Author Response · Author response to Decision Letter 0]

4 Nov 2023

Included in the response to the reviewers letter.

---

## [Decision Letter · Decision Letter 1]

14 Dec 2023

PONE-D-23-14246R1Traditional Herbal Medicine Use Doubled the Risk of Multi-organ Dysfunction Syndrome in a Pediatric Intensive Care Unit in the University of Gondar Comprehensive Specialized Hospital, Northwest Ethiopia: A Prospective Cohort Study.PLOS ONE

Dear Dr. Teshager,

Thank you for submitting your manuscript to PLOS ONE. After careful consideration, we feel that it has merit but does not fully meet PLOS ONE’s publication criteria as it currently stands. Therefore, we invite you to submit a revised version of the manuscript that addresses the points raised during the review process.

Please revise your Title: 1. shorten it; 2. remove the institution name.

The Title is too long.  Please revise it.

We look forward to receiving your revised manuscript.

Kind regards,

Academic Editor

PLOS ONE

Journal Requirements:

Reviewers' comments:

Reviewer's Responses to Questions

**Comments to the Author**

1. If the authors have adequately addressed your comments raised in a previous round of review and you feel that this manuscript is now acceptable for publication, you may indicate that here to bypass the “Comments to the Author” section, enter your conflict of interest statement in the “Confidential to Editor” section, and submit your "Accept" recommendation.

Reviewer #1: All comments have been addressed

Reviewer #2: All comments have been addressed

2. Is the manuscript technically sound, and do the data support the conclusions?

Reviewer #1: Yes

Reviewer #2: Yes

3. Has the statistical analysis been performed appropriately and rigorously? 

Reviewer #1: Yes

Reviewer #2: Yes

4. Have the authors made all data underlying the findings in their manuscript fully available?

Reviewer #1: Yes

Reviewer #2: Yes

5. Is the manuscript presented in an intelligible fashion and written in standard English?

Reviewer #1: Yes

Reviewer #2: Yes

6. Review Comments to the Author

Reviewer #1: The manuscript is technically sound and written in standard English. They used appropriate statistical analysis and concluded based on their core findings. The declare that the authors addressed all the comments and revised it accordingly. However, I have seen few inconsistency and editorial issue on certain points and I recommend the authors/editors to see carefully those areas.

1. In the abstract section before the last sentence, "A threefold increase in the risk of MODS....) has to be stated consistently as " A more than twofold increase in the risk of MODS was seen in patients who used TMH".

2. In the competing interest section, it is stated that "---patients/public were not involved in the research". However, the study is a prospective cohort and needs patients follow-up. How it is said that patients were not involved?

3. In the data processing and analysis section, the first sentence has to be edited as of the following. "After ensuring consistency and completeness, the data was imported into EpiData V.3.1, then exported to Excel and subsequently to R 4.2.2 for cleaning and analysis."

4. In the data processing and analysis section, the authors has to replace the term multivariate by Multivariable, since the study has only one dichotomous outcome.

Reviewer #2: I hereby endorse the publication of the manuscript titled “Traditional Herbal Medicine Use Doubled the Risk of Multi-organ Dysfunction Syndrome in a Pediatric Intensive Care Unit in the University of Gondar Comprehensive Specialized Hospital, Northwest Ethiopia: A Prospective Cohort Study”. I believe that the findings of this study provide valuable insights into the association between traditional herbal medicine use and the risk of Multi-organ Dysfunction Syndrome in pediatric patients.

7. PLOS authors have the option to publish the peer review history of their article (what does this mean?). If published, this will include your full peer review and any attached files.

Reviewer #1: **Yes: **Dr. Dawit Kassaye Getaneh

Reviewer #2: No

---

## [Author Response · Author response to Decision Letter 1]

25 Dec 2023

1. The Title is too long. Please revise it.

Response: Thank you for the suggestion. The title is revised. I ,previously, included the institution name as per the recommendation of one of the reviewers. 

2. Please review your reference list to ensure that it is complete and correct.

Response: we have reviewed and made sure the references are correct.

Response to Reviewer #1

1. In the abstract section before the last sentence, "A threefold increase in the risk of MODS....) has to be stated consistently as " A more than twofold increase in the risk of MODS was seen in patients who used TMH". 

Response: Thank you, reviewer, for the input. We have corrected it in the revised manuscript as to your recommendation. 

2. In the competing interest section, it is stated that "---patients/public were not involved in the research". However, the study is a prospective cohort and needs patient’s follow-up. How it is said that patients were not involved?

Response: Thank you for bringing to my attention the apparent contradiction in the competing interest section. By "---patients/public were not involved in the research “, we mean that patients/public are not directly involved in the design and data analysis of the research. We have modified it as “There were no competing interests, patients and/or public were not directly involved in the design, conduct, or reporting of the research.” 

3. In the data processing and analysis section, the first sentence has to be edited as of the following. "After ensuring consistency and completeness, the data was imported into EpiData V.3.1, then exported to Excel and subsequently to R 4.2.2 for cleaning and analysis." 

Response: Thank you, reviewer. Done as to your recommendation. 

4. In the data processing and analysis section, the authors has to replace the term multivariate by Multivariable, since the study has only one dichotomous outcome.

Response: Thank you, reviewer. Corrected as to your recommendation.

---

## [Decision Letter · Decision Letter 2]

16 Jan 2024

PONE-D-23-14246R2Traditional Herbal Medicine Use Doubled the Risk of Multi-organ Dysfunction Syndrome in Children in Gondar, Ethiopia: A Prospective Cohort Study.PLOS ONE

Dear Dr. Teshager,

Thank you for submitting your manuscript to PLOS ONE. After careful consideration, we feel that it has merit but does not fully meet PLOS ONE’s publication criteria as it currently stands. Therefore, we invite you to submit a revised version of the manuscript that addresses the points raised during the review process.

Please revise the Title.  Is it necessary to specify the location in the Title?

We look forward to receiving your revised manuscript.

Kind regards,

Academic Editor

PLOS ONE

Journal Requirements:

Reviewers' comments:

Reviewer's Responses to Questions

**Comments to the Author**

1. If the authors have adequately addressed your comments raised in a previous round of review and you feel that this manuscript is now acceptable for publication, you may indicate that here to bypass the “Comments to the Author” section, enter your conflict of interest statement in the “Confidential to Editor” section, and submit your "Accept" recommendation.

Reviewer #1: All comments have been addressed

Reviewer #2: All comments have been addressed

2. Is the manuscript technically sound, and do the data support the conclusions?

Reviewer #1: Yes

Reviewer #2: Yes

3. Has the statistical analysis been performed appropriately and rigorously? 

Reviewer #1: Yes

Reviewer #2: Yes

4. Have the authors made all data underlying the findings in their manuscript fully available?

Reviewer #1: Yes

Reviewer #2: Yes

5. Is the manuscript presented in an intelligible fashion and written in standard English?

Reviewer #1: Yes

Reviewer #2: Yes

6. Review Comments to the Author

Reviewer #1: I found it well organized and technically sound research paper. All my concerns are addressed properly and it is well edited.

Reviewer #2: Re-review comment of Reviewer 2

I believe that the findings of this study offer valuable insights into the association between the use of traditional herbal medicine and the risk of Multi-organ Dysfunction Syndrome in pediatric patients. However, I do have some minor comments.

1. In the Data processing and analysis section, the statement "Statistical tests such as the chi-square test and simple linear regression were used when they are found to be necessary" should be removed. These tests are not appropriate for survival data analysis in this study, and the statement is inaccurate. Instead, please include only the methods that were actually employed for analysis.

2. The section currently titled “Predictors of Mortality in the PICU” should be renamed to “Predictors of Incidence of MODS in the PICU”. Furthermore, all instances of the term “mortality” should be replaced with “incidence of MODS” in this section. This is because the primary objective of this study is to investigate the incidence and predictors of MODS in a PICU, not mortality. Additionally, please note that Table 2 does not discuss mortality.”

3. In the Discussion section, the statement “The finding that patients with critical illness diagnoses had a 2.45 times increased risk of developing MODS in critically ill patients is supported by multiple other studies” should be corrected. The correct value is not 2.45, but rather 2.68.

4. In the Author Contributions section, the roles of the authors need to be clearly defined.

7. PLOS authors have the option to publish the peer review history of their article (what does this mean?). If published, this will include your full peer review and any attached files.

Reviewer #1: **Yes: **Dr. Dawit Kassaye Getaneh (DVM. BPharm, MPH)

Reviewer #2: **Yes: **Leykun Getaneh Gebeye

---

## [Author Response · Author response to Decision Letter 2]

29 Jan 2024

Response to Editorial Board:

Dear editorial Boar, 

We have amended the manuscript as to the recommendations and comments you provided us. Thank you for contributing for the improvement of our work. We hope we have addressed all the issues/points you stated. 

1. Is it necessary to specify the location in the Title?

Response: Thank you for the suggestion. The title is revised. 

2. Journal Requirements: References

Response: I have checked my citations. All the articles I cited are available online. I checked them one by one on google scholar. 

Response to Reviewer #2

1. In the Data processing and analysis section, the statement "Statistical tests such as the chi-square test and simple linear regression were used when they are found to be necessary" should be removed. These tests are not appropriate for survival data analysis in this study, and the statement is inaccurate. Instead, please include only the methods that were actually employed for analysis

Response: Thank you, reviewer, for the input. We have corrected it in the revised manuscript as to your recommendation. As this is survival analysis, we used KM curve, log rank test and Cox PH model. 

2. The section currently titled “Predictors of Mortality in the PICU” should be renamed to “Predictors of Incidence of MODS in the PICU”. Furthermore, all instances of the term “mortality” should be replaced with “incidence of MODS” in this section. This is because the primary objective of this study is to investigate the incidence and predictors of MODS in a PICU, not mortality. Additionally, please note that Table 2 does not discuss mortality.”

Response: Thank you, reviewer, that was unseen mistake on our side. We have corrected it in the revised manuscript as to your recommendation. Regarding table 2, it is a regression table for explanatory variables of MODS. On the footnote, we put the long form of mPIM 2 : modified pediatric mortality score 2 which we used to assess severity of illness at admission. 

3. In the Discussion section, the statement “The finding that patients with critical illness diagnoses had a 2.45 times increased risk of developing MODS in critically ill patients is supported by multiple other studies” should be corrected. The correct value is not 2.45, but rather 2.68.

Response: Thank you. Corrected in the revised manuscript. 

4. In the Author Contributions section, the roles of the authors need to be clearly defined.

Response: Thank you, reviewer. We have clearly put the authors’ contribution in the revised manuscript

---

## [Decision Letter · Decision Letter 3]

9 Feb 2024

Traditional Herbal Medicine Use Doubled the Risk of Multi-organ Dysfunction Syndrome in Children: A Prospective Cohort Study.

PONE-D-23-14246R3

Dear Dr. Teshager,

We’re pleased to inform you that your manuscript has been judged scientifically suitable for publication and will be formally accepted for publication once it meets all outstanding technical requirements.

Kind regards,

Academic Editor

PLOS ONE

Additional Editor Comments (optional):

Reviewers' comments:

Reviewer's Responses to Questions

**Comments to the Author**

1. If the authors have adequately addressed your comments raised in a previous round of review and you feel that this manuscript is now acceptable for publication, you may indicate that here to bypass the “Comments to the Author” section, enter your conflict of interest statement in the “Confidential to Editor” section, and submit your "Accept" recommendation.

Reviewer #1: All comments have been addressed

Reviewer #2: All comments have been addressed

2. Is the manuscript technically sound, and do the data support the conclusions?

Reviewer #1: Yes

Reviewer #2: Yes

3. Has the statistical analysis been performed appropriately and rigorously? 

Reviewer #1: Yes

Reviewer #2: Yes

4. Have the authors made all data underlying the findings in their manuscript fully available?

Reviewer #1: Yes

Reviewer #2: Yes

5. Is the manuscript presented in an intelligible fashion and written in standard English?

Reviewer #1: Yes

Reviewer #2: Yes

6. Review Comments to the Author

Reviewer #1: The manuscript is technically sound, well edited and presented following a scientific methods using a standard English.

Reviewer #2: The manuscript titled “Traditional Herbal Medicine Use Doubled the Risk of Multi-organ Dysfunction Syndrome in Children: A Prospective Cohort Study” is approved for publication. The revisions have enhanced its quality, and no further changes are required.

7. PLOS authors have the option to publish the peer review history of their article (what does this mean?). If published, this will include your full peer review and any attached files.

Reviewer #1: **Yes: **Dr. Dawit Kassaye Getaneh (MPH, Assist. professor)

Reviewer #2: **Yes: **Leykun Getaneh Gebeye

---

## [Editor Report · Acceptance letter]

14 Feb 2024

PONE-D-23-14246R3 

PLOS ONE

Dear Dr. Teshager, 

I'm pleased to inform you that your manuscript has been deemed suitable for publication in PLOS ONE. Congratulations! Your manuscript is now being handed over to our production team.

Kind regards, 

on behalf of

Dr. Robert Jeenchen Chen 

Academic Editor

PLOS ONE